| Editor's Pick | Environmental Microbiology | Research Article

# Increasing transposase abundance with ocean depth correlates with a particle-associated lifestyle

Juntao Zhong,[1,2] Troy Osborn,[1] Thais Del Rosario Hernández,[1,3] Oleksandr Kyrysyuk,[1,4] Benjamin J. Tully,[5] Rika E. Anderson[1]

**ABSTRACT**    Transposases are mobile genetic elements that move within and between genomes, promoting genomic plasticity in microorganisms. In marine microbial communities, the abundance of transposases increases with depth, but the reasons behind this trend remain unclear. Our analysis of metagenomes from the *Tara* Oceans and Malaspina Expeditions suggests that a particle-associated lifestyle is the main covariate for the high occurrence of transposases in the deep ocean, and this trend holds true for individual genomes as well as in a community-wide sense. We observed a strong and depth-independent correlation between transposase abundance and the presence of biofilm-associated genes, as well as the prevalence of secretory enzymes. This suggests that mobile genetic elements readily propagate among microbial communities within crowded biofilms. Furthermore, we show that particle association positively correlates with larger genome size, which is in turn associated with higher transposase abundance. Cassette sequences associated with transposons are enriched with genes related to defense mechanisms, which are more highly expressed in the deep sea. Thus, while transposons spread at the expense of their microbial hosts, they also introduce novel genes and potentially benefit the hosts in helping to compete for limited resources. Overall, our results suggest a new understanding of deep ocean particles as highways for gene sharing among defensively oriented microbial genomes.

**IMPORTANCE**    Genes can move within and between microbial genomes via mobile genetic elements, which include transposases and transposons. In the oceans, there is a puzzling increase in transposase abundance in microbial genomes as depth increases. To gain insight into this trend, we conducted an extensive analysis of marine microbial metagenomes and metatranscriptomes. We found a significant correlation between transposase abundance and a particle-associated lifestyle among marine microbes at both the metagenome and genome-resolved levels. We also observed a link between transposase abundance and genes related to defense mechanisms. These results suggest that as microbes become densely packed into crowded particles, mobile genes are more likely to spread and carry genetic material that provides a competitive advantage in crowded habitats. This may enable deep sea microbes to effectively compete in such environments.

**KEYWORDS**    transposase, marine microbiology

Mobile genetic elements (MGEs) are segments of DNA that facilitate the movement of genetic sequences within and between bacterial and archaeal genomes (1). MGEs generally encode enzymes to mediate this process (1): in transposons, the mediating enzymes are transposases, and the mechanism of transfer results in inverted repeats (2). During migration, some MGEs carry cassette sequences that often contain functional genes, including those for antibiotic resistance (3), metal uptake (4, 5), and regulatory genes influencing gene expression in host cells (6, 7).

Address correspondence to Rika E. Anderson, randerson@carleton.edu.

The authors declare no conflict of interest.

See the funding table on p. 16.

Transposases are the most abundant and ubiquitous genes in nature (8), although their distribution among taxa is uneven (9, 10). In marine systems, one of the most striking—and as of yet unexplained—trends in microbial genomics is the distinct increase in transposase abundance with depth. Genomic analyses of samples collected at station ALOHA in the North Pacific revealed a substantial increase in transposase abundance from 500 m to their observed maximum at 4,000 m (11, 12). Transposases were one of the most overrepresented cluster of orthologous gene (COG) categories in ALOHA deep waters, accounting for 1.2% of all fosmid sequences from 4,000 m (11, 13). Similarly, a study of hydrothermal chimneys demonstrated a high abundance of transposases in biofilms, comprising 8% of all metagenomic reads, which is 10 times higher than observed in metagenomes from other habitats (14). Based on these observations, our analyses focused on three central questions: (i) why does transposase abundance increase with depth in marine systems? (ii) are transposases selfish genes with neutral or deleterious effects on host genomes? and (iii) do transposases provide useful functions to the microbial hosts that harbor them?

To better understand the high transposase abundance in the deep sea and to gain insights into the role of MGEs in marine microbial communities, we analyzed 138 microbial metagenomes and 152 microbial metatranscriptomes from the *Tara* Oceans Expedition (15, 16). The *Tara* Oceans samples spanned depths from 5 m to 1,000 m. In order to represent bathypelagic microbial communities, we also incorporated 58 metagenomes from the 2010 Malaspina Expedition (17), collected between 2,400 m and 4,000 m. To explore the role of transposases at a genome-resolved level, we analyzed a total of 3,290 metagenome-assembled genomes (MAGs) previously generated from the *Tara* Oceans (18, 19, 20) and Malaspina (17) metagenomes. Previous studies have shown that deep-sea prokaryotes have a predominantly particle-associated lifestyle (21). Here, we show that the increasing abundance of transposases with depth in ocean microbial communities is associated with a shift toward an inferred particle-associated lifestyle. In addition to particle association, we identified taxonomy and genome size as key covariates with transposase abundance in MAGs. Additionally, we observed a high abundance of open reading frames (ORFs) in the functional category "defense mechanisms" among the ORFs in cassette sequences associated with transposases, suggesting that transposons introduce beneficial genes to their microbial hosts inhabiting highly competitive habitats.

## RESULTS AND DISCUSSION

Previous studies have revealed that transposase abundance in microbial genomes is associated with increased ocean depth, lower dissolved oxygen (DO) concentrations, and a particle-associated lifestyle (as opposed to a planktonic lifestyle) (9, 12, 22). However, the reason and nature of these correlations have not been further explored. To confirm these findings on a broader scale, we screened for significant covariates for transposase abundance in the *Tara* Oceans and Malaspina metagenomes. We plotted the transposase abundance against depth and DO. In our metagenome analysis, transposase abundance was defined as the proportion of reads mapped back to all transposase ORFs (the database used to identify putative transposase ORFs is shown in Table S1). We confirmed that transposase abundance increases steadily as depth increases, despite large variations across samples (Fig. 1). In every ocean, the transposase abundances of the deep water samples were higher than those of the shallow water samples (Table S2). Here, "deep water" was defined as mesopelagic (depth 250 m–1,000 m) and bathypelagic (2,500–4,000 m) waters, and "shallow water" was defined as surface (<10 m) and deep chlorophyll maximum (DCM, depth 17 m–120 m) waters.

To investigate how MGE abundances relate to the planktonic and particle-associated lifestyle, we calculated transposase abundance in metagenomes collected from different filter sizes. In the *Tara* Oceans samples, samples from the 0.22 μm–3.0 μm filter size fraction were considered to be enriched in particle-associated cells, and samples from the 0.22 μm–1.6 μm size fraction were considered to be enriched in planktonic cells. We

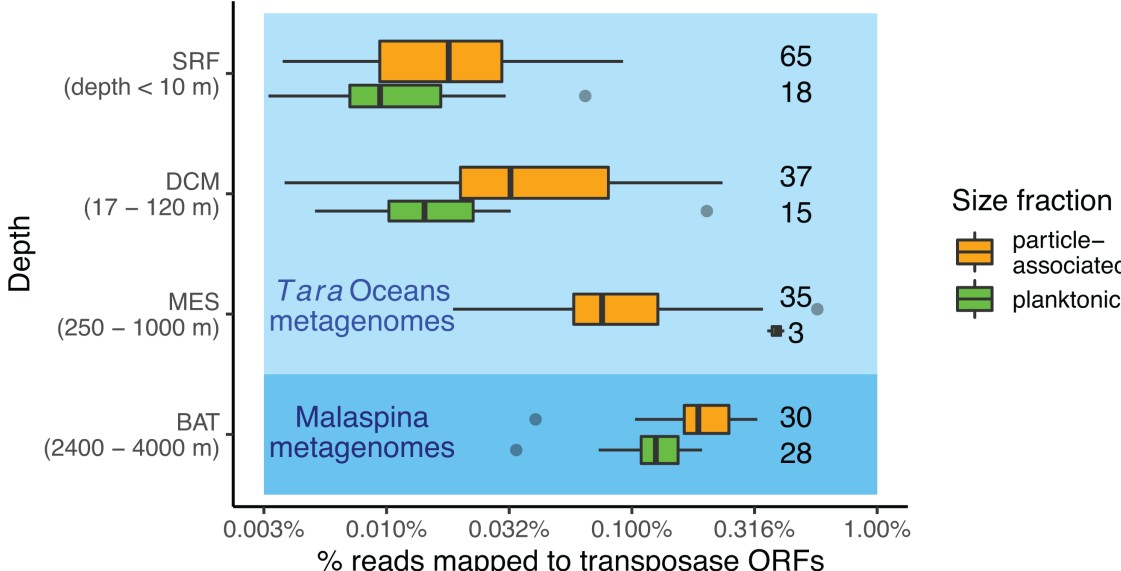

**FIG 1** Samples from the deep ocean have higher transposase abundance. The abundance of transposase ORFs in metagenomic samples from the *Tara* Oceans and Malaspina Expedition, separated by depth and by filter size fraction. SRF, surface; DCM, deep chlorophyll maximum; MES, mesopelagic zone; BAT, bathypelagic zone. For *Tara* Oceans samples, size fractions were between 0.22 µm and 3.0 µm (here called "particle associated") and 0.22 µm and 1.6 µm (here called "planktonic"). For Malaspina samples, the "particle-associated" size fraction was between 0.8 µm and 20 µm, and the "planktonic" size fraction was between 0.2 µm and 0.8 µm. The widths of the boxplots reflect the sample count in each category (counts shown on the right).

focused on these two size fractions for full metagenomic analysis as they are considered to be prokaryote-enriched (16); larger size fractions were considered to be eukaryote-enriched and thus were not included for full metagenomic analysis. In the Malaspina samples, samples from the 0.8 µm–5.0 µm size fraction were considered to be particle-associated and 0.2 µm–0.8 µm were planktonic (18). After controlling for depth, samples from particle-associated microbial communities were more enriched in transposases than their planktonic counterparts (Fig. 1; analysis of variance [ANOVA] $F$-test $P = 0.0032$).

The depth of a sample (e.g., surface, DCM) was a significant predictor for transposase abundance across all prokaryote-enriched size fractions (Table S3). Adding temperature, however, did not significantly improve the prediction of transposase abundance in a sample (ANOVA $F$-test $P = 0.48$, Table S3; Fig. S1A). Similarly, adding dissolved oxygen also did not improve the prediction of transposase abundance (ANOVA $F$-test $P = 0.09$, Table S3). This was partially due to the inconsistent relationship between DO and transposase abundance across depths. In the surface and bathypelagic samples, DO positively correlated with transposase abundance, but this correlation was reversed in DCM and mesopelagic samples (Fig. S1B).

## Transposase abundance is positively correlated with a particle-associated lifestyle in all depths

To further investigate the relationship between transposase abundance and particle-associated lifestyles, we examined the correlation between transposase abundance and the percentage of predicted secretory carbohydrate-active enzymes (CAZymes) and peptidases among all CAZyme and peptidase ORFs (see Materials and Methods). We conducted these analyses across all prokaryote-enriched metagenomic samples. The percentage of secretory enzymes among all CAZymes and peptidases has previously been applied to quantify the degree to which microbial communities rely on a particle-associated lifestyle (21). Microorganisms rely on extracellular enzymes to degrade large particulate organic carbon into compounds of smaller molecular weight to incorporate into the cell (21, 23) and CAZymes and peptidases are key enzymes for carbohydrate and protein degradation, respectively (24, 25). To account for differences in biological

processes at different depths, we normalized the gene and transcript abundance of predicted secretory CAZymes and peptidases by those of all CAZyme and peptidase ORFs in metatranscriptomic and metagenomic samples (the bathypelagic zone was excluded for all transcript analyses, because no metatranscriptomes were available for Malaspina samples). The large number of CAZymes and peptidases in the database make it challenging to identify their exact binding substrates, which might negatively impact our ability to quantify the particle association of a microbial community. However, microorganisms often use biofilms to attach to the surfaces of particles (23), and thus, we paired the CAZyme/peptidase analysis with a secondary analysis quantifying the abundance of biofilm-associated genes using a manually curated database of biofilm-associated ORFs (see Table S4 for accession numbers for the database of biofilm-associated genes).

From the *Tara* Oceans metagenomes, we identified 52,890 secretory CAZymes out of 421,080 CAZyme ORFs, and 179,932 secretory peptidases out of 1,294,210 peptidase ORFs; from the Malaspina metagenomes, we identified 7,002 secretory CAZymes in 40,415 CAZyme ORFs, and 18,014 secretory peptidases out of 67,617 peptidases (see Materials and Methods). We observed an increasing percentage of secretory CAZymes and peptidases with depth in both metagenomic and metatranscriptomic samples (Fig. S2A through D), which was consistent with previous findings (21). The biofilm-associated ORFs showed a similar increase in gene and transcript abundance with depth (Fig. S2E and F). The increase in the percentage of secretory CAZymes and peptidases with depth indicates increased extracellular enzyme activity, suggesting a shift toward a particle-associated lifestyle toward the deep ocean.

Marine particles are thought to be hotspots for horizontal gene transfer and transposase propagation (25), and the increased importance of a particle-associated lifestyle with depth in the ocean offers a promising explanation for the elevated transposase abundance in the deep ocean. In both metagenomic and metatranscriptomic samples, the abundance of transposases and the percentage of secretory CAZymes and peptidases were strongly correlated in general as well as within each depth (Fig. 2). Given a model with only depth as predictor of the transposase abundance in a sample, the addition of secretory CAZymes and peptidases each significantly improved the accuracy of prediction (both ANOVA $F$-test $P = 3 \times 10^{-7}$, Table S3). The depth-independent association between a particle-associated lifestyle and transposase abundance was also supported by persistent correlations between the abundance of transposase genes/transcripts with those of biofilm-associated ORFs in each depth (Fig. S3). Correlations in metagenomes and metatranscriptomes suggested that transposases are more abundant and more frequently transcribed in microbial communities with a greater reliance on a particle-associated lifestyle. After observing such trends at the community level, we sought to determine if these trends hold true at a genome-resolved level to support our hypothesis that the particle-associated lifestyle is a main driver for the high transposase abundance in the deep ocean.

## Genomes from particle-associated samples have elevated transposase abundance

In order to determine whether the patterns identified above were consistent at the genome level, we undertook an analysis of 1,888 MAGs synthesized from co-assemblies of the *Tara* Oceans data that had previously been characterized across distinct size fractions (20). One major advantage to this approach was that transposases could be quantified on a genome-by-genome basis, and each genome could be designated as particle-associated or not based on their relative abundance in metagenomes from different size fractions. The 1,888 MAGs we used for this analysis were mostly bacterial ($n$ = 1,778) with the rest being archaeal ($n$ = 110), and all MAGs exhibited >70% completion and an average redundancy of 2.5%. From this analysis, we found that MAGs from the smallest size fraction (0.22 µm–5 µm), which should largely consist of planktonic cells, contained fewer transposases than MAGs from each of the larger size fractions, each of

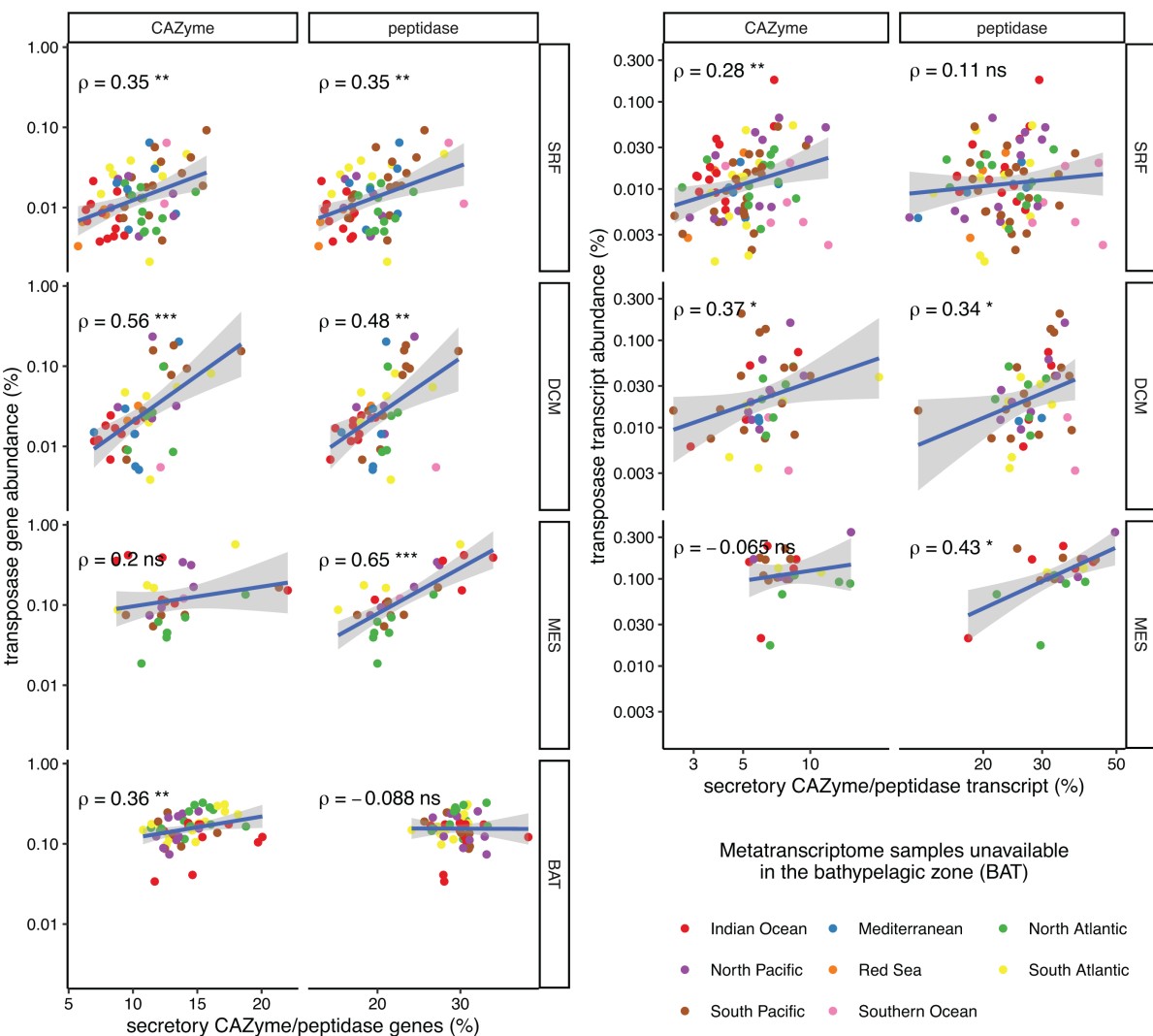

**FIG 2** The gene potential (DNA) and transcript abundance (RNA) of transposases correlates with those of secretory CAZymes and peptidases. (A) The correlation between the abundance of transposases and secretory CAZyme and peptidases in metagenomes, separated according to marine layer. (B) The same correlation in metatranscriptomes: colors represent different ocean depths. The Spearman's correlation coefficients, ρ, are shown on the top left. *P < 0.05, **P < 0.01, ***P < 0.001, ****P < 0.0001. Note that the percent transcript abundances are log-transformed.

which should largely consist of particle-associated cells (Fig. 3). In particular, the proportion of MAGs with no transposases at all was found to be significantly different in the smallest size fraction compared to the larger size fractions ($\chi^2$-test, $P < 2.2e-16$). On the other hand, the proportion of MAGs with no transposases was not found to be significantly different among the larger size fractions ($\chi^2$-test, $P = 0.8156$), as would be expected if the MAGs from the smallest size fraction were largely planktonic and if MAGs from the larger size fractions were largely particle-associated. Taken together, these insights provide strong support for the hypothesis that transposase abundance is linked to a particle-associated lifestyle.

## Genome size and taxonomy are key factors to transposase abundance in populations

Given the enrichment of transposase ORFs in particle-associated MAGs, we ask three questions: (i) whether the correlation between transposase abundance persists with depth in MAGs; (ii) since larger genomes have an elevated rate of horizontal gene transfer (HGT) (24, 26), whether genome size correlates to transposase abundance in

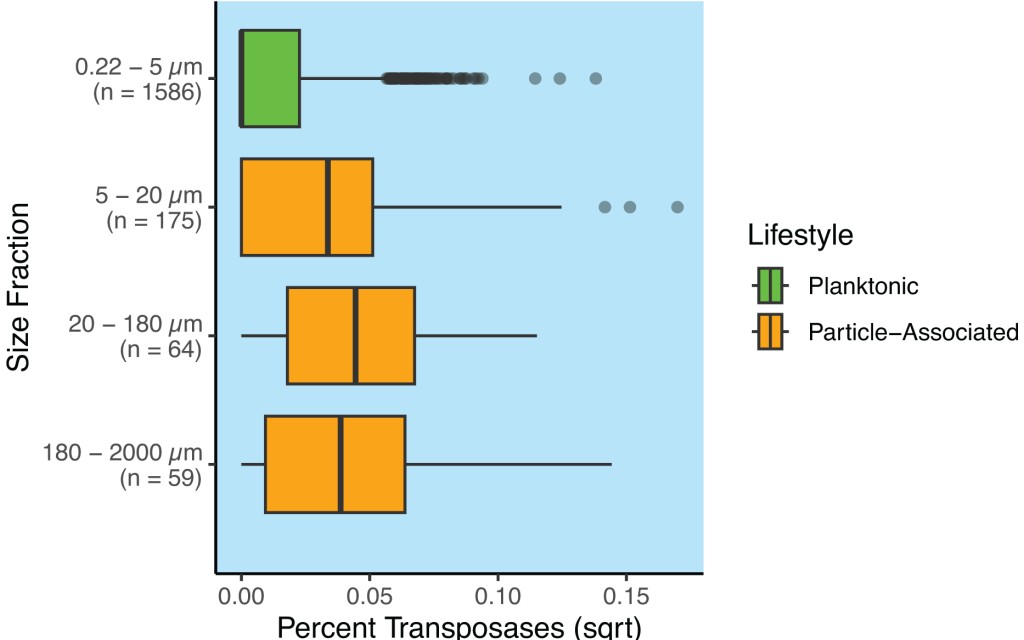

**FIG 3** Transposase abundance is markedly higher for particle-associated MAGs than planktonic ones. The *x*-axis here is square root-transformed for clarity, as log-transforming was not an option due to the amount of zeros in the data. Close to 70% of planktonic MAGs contained no transposases at all—a proportion that was found to be significantly different from the proportion of MAGs containing no transposases in the particle-associated group (see Results and Discussion). For this analysis, MAGs were considered planktonic if they came from the smallest size fraction (0.22 μm–5 μm) and particle-associated otherwise, as denoted by the colors of the boxplots. Note the wide difference in sample sizes among the size fractions.

MAGs; and (iii) whether specific taxa encode more transposases than others, and thus whether community composition plays a role in determining transposase abundance. For this set of genome-resolved analyses, we analyzed a different set of 1,147 MAGs recovered from the *Tara* Oceans metagenomes across different depths and size fractions (18) as well as 255 MAGs from the Malaspina metagenomes (17). We chose to analyze this set of MAGs for these analyses because they spanned a wider depth range than the MAGs described in the analysis above. All MAGs in this analysis had percent completeness >70%, redundancy <10%, and contained <5% of eukaryotic sequences (see Materials and Methods). The transposase abundance in MAGs was quantified by the percentage of transposase ORFs among all ORFs in the MAG (abbreviated as %-transposase).

To address the first question, we found that MAGs from deep waters had higher transposase abundance than MAGs from shallow waters. Specifically, MAGs from the mesopelagic and bathypelagic zones had a mean %-transposase six times (95% CI: 5.38 to 6.57) that of MAGs from the surface and DCM.

Given that this set of MAGs was not classified by size fraction, we quantified the degree to which individual genomes were particle-associated by calculating the percentage of secretory CAZymes and peptidases out of all CAZyme and peptidase ORFs (a ratio of counts) in a MAG. This method has been verified in metagenomes separated through serial filtering techniques—MAGs assembled from samples with larger filter sizes contain a higher percentage of secretory CAZymes and peptidases in the genome compared to those filtered with smaller size fractions (27). We also independently validated this method by comparing the percentage of secretory CAZyme and peptidase ORFs in taxa generally known to be particle-associated or planktonic (Fig. S4). Taxonomy was analyzed at a class level, and the "complete genome size" was estimated using the genome length of a MAG divided by its estimated percent completeness. Although MAG assembly might bias against MGEs (28) and the completeness of MAGs might be

underestimated for rare taxonomy groups (29, 30), we can still gain valuable insights by analyzing the distribution of transposase ORFs in MAGs of various genome sizes. To address the first question, we observed a positive correlation between the percentage of transposase ORFs and the percentage of secretory CAZymes and peptidases in MAGs (Spearman $\rho = 0.208$ and $\rho = 0.210$, respectively; both $P < 10^{-14}$), thus confirming that the correlation between transposases and particle association persists on a genome-by-genome basis.

To examine the relationship between genome size and transposase abundance, we compared the estimated genome size of each MAG with the percentage of ORFs characterized as transposases within each MAG. In every depth, we observed a positive correlation between the estimated genome size and the percentage of transposase ORFs within a MAG (Spearman $\rho$ was 0.56, 0.50, 0.46, and 0.60 for surface, DCM, mesopelagic, and bathypelagic samples, respectively; all $P < 10^{-20}$), confirming previous speculations that larger genomes were linked to high transposase abundance (9, 12). While deep water genomes were bigger (Fig. 4A), the particle-associated lifestyle correlated with greater complete genome size in every depth (Fig. 5B). Previous work has shown that large genomes are associated with a high fraction of transposase ORFs in isolated microbial genomes (10, 31, 32), and in a study of metagenomes from the Baltic Sea (9). The correlation between transposase abundance and genome size was previously attributed to a higher frequency of HGT in larger genomes (24, 31). A correlation between the particle-associated lifestyle and genome size has also been previously observed and may be linked to the expanded metabolic versatility of particle-associated microbial populations (27). Thus, the particle-associated lifestyle might facilitate transposase propagation by reducing the distance between microorganisms, and may lead to larger genome size through increased rates of HGT. It is worth noting that the transposase abundance in MAGs peaked when the complete genome size reached 5 Mbp, but it stabilized or declined beyond the peak (Fig. 4C). A previous study has also reported a decreased proportion of transposase ORFs after bacterial genomes surpass 6 Mbp in size (10). However, most (88.9%) MAGs recovered from the *Tara* Oceans and Malaspina metagenomes were under 5 Mbp, so the association between large genomes and high transposase abundance generally held true.

To address the third question, we found that taxonomic class was also a key predictor of transposase abundance in MAGs; information on the taxonomy and depth of MAGs together explained 53% of the variance in transposase abundance (Table 1). MAGs of the taxonomic classes *Alphaproteobacteria*, *Betaproteobacteria*, *Gammaproteobacteria*, and *Actinobacteria* were enriched in transposases compared to other MAGs (for each class with ≥10 MAGs, the %-transposase of MAGs in that class was compared against those of all other MAGs; Wilcoxon test *P* cutoff: 0.05). MAGs of *Flavobacteria*, *Acidimicrobidae*, *novelClass_E*, and *SAR202-2* had low transposase abundance. Taxa that were enriched/low in transposases mostly matched with previous studies (9, 10), except that *Actinobacteria* had previously been reported to be low in transposase abundance.

The link between taxonomy and transposase abundance could partially be attributed to depth. MAGs of transposase-enriched classes were more abundant in deep waters, and MAGs from low transposase classes were more abundant in shallow waters (2 $\chi^2$-tests, both $P < 1 \times 10^{-6}$; Fig. S5). Thus, although community composition is a key covariate to transposase abundance, the distribution of taxa is nonetheless linked to the depth of a microbial community.

## Relaxed selection does not explain transposase abundance in genomes

It is possible that transposases accumulate in deep ocean microbial populations as a result of genetic drift, because those populations tend to be smaller in size (33) and experience slow growth rates (34). To examine this possibility, we tested whether the strength of selection experienced by populations correlated with their transposase abundance. For each ORF in each MAG, we calculated its *pN/pS* ratio in each sample from its designated depth (*pN/pS* was only calculated for ORFs with ≥20× coverage).

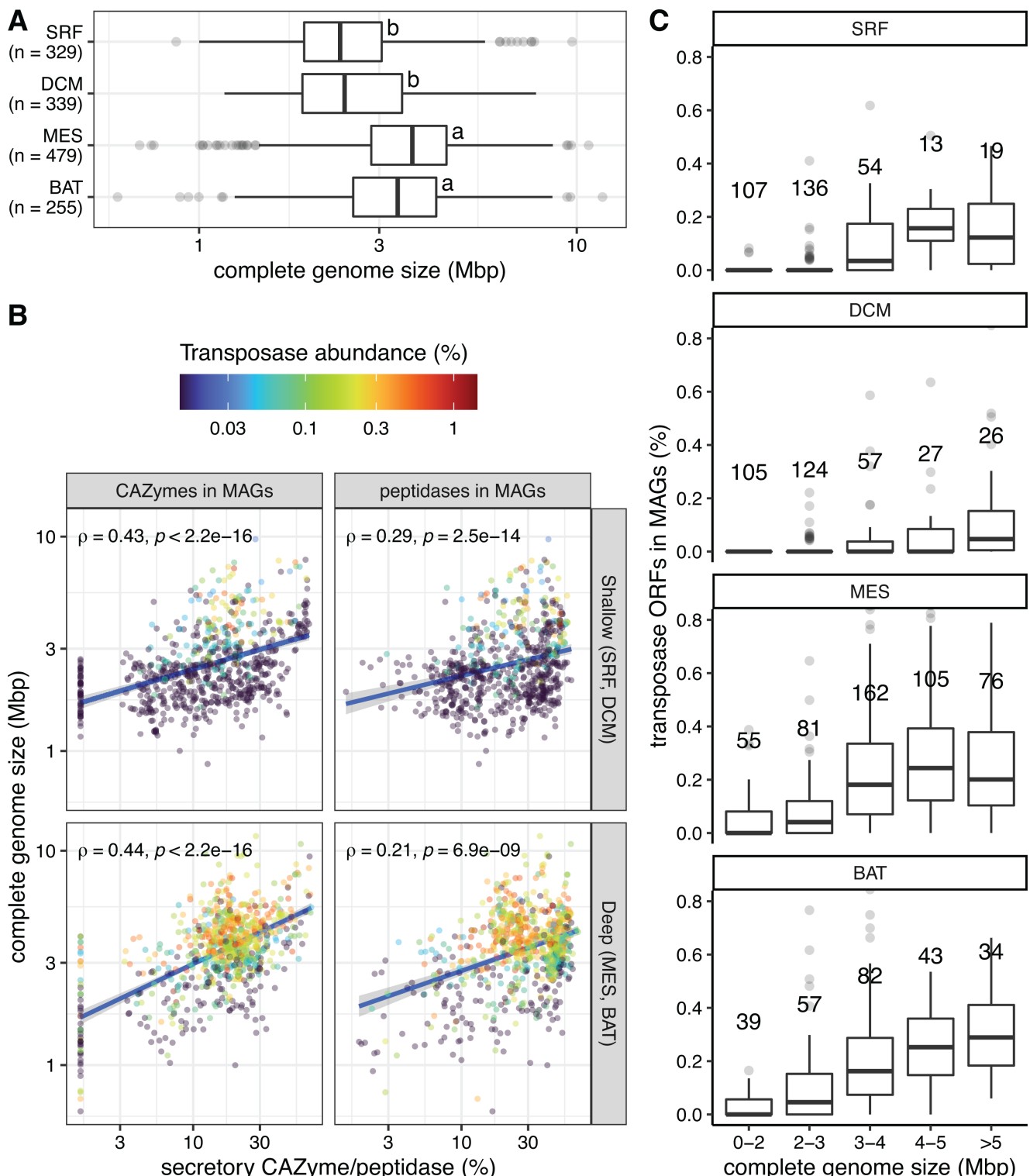

**FIG 4** Deep-sea and particle-associated microbial populations tend to have larger genomes, which correlate with high transposase abundance in those populations. (A) The estimated complete genome size of MAGs (complete genome size = number of base pairs in a MAG/% completeness), grouped by depth. Letters on boxplots were generated from the Tukey honestly significant difference test. (B) Scatterplots showing the correlation between secretory CAZymes/peptidases and the estimated complete genome size of MAGs. (C) The transposase abundance in MAGs, grouped by depth and complete genome size. See Materials and Methods for the determination of depth of a MAG.

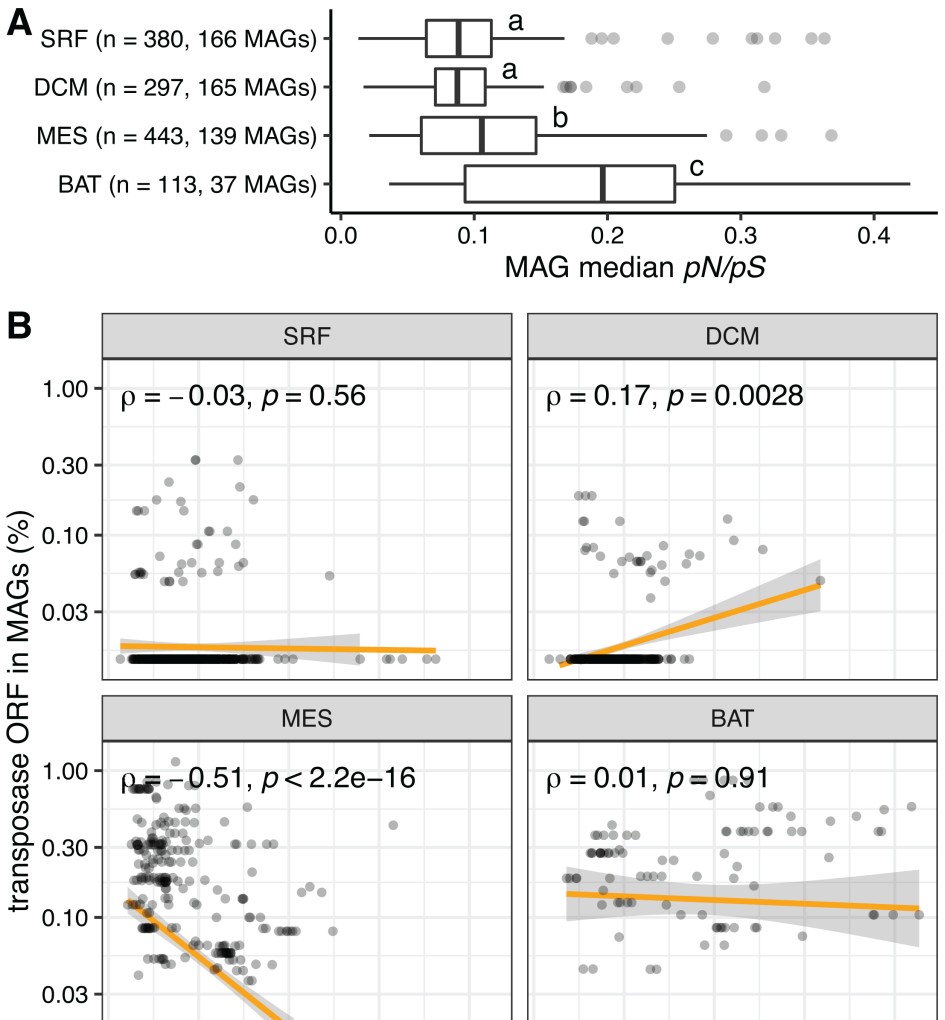

**FIG 5** Selection pressure and transposase abundance are not correlated on a genome-resolved scale. (A) The median *pN/pS* ratio of MAGs, grouped by depth. Since *pN/pS* were calculated on a per-sample basis, a MAG might have several median *pN/pS* ratios from multiple samples. (B) The relationship between the selection pressure (median *pN/pS*) and the transposase abundance of a MAG, separated by depth. Only MAGs with ≥100 *pN/pS* were computed for median *pN/pS*.

*pN*/pS represents the proportion of nonsynonymous mutations to the proportion of synonymous mutations, and characterizes selection at the level of the population, in contrast to *dN/dS*, which characterizes selection between individual species (35–37).

We observed that MAGs from the mesopelagic zone had a higher median *pN/pS* than MAGs from the surface and DCM, and MAGs from the bathypelagic zone had a higher median *pN/pS* than MAGs from the mesopelagic zone (both $P < 5 \times 10^{-6}$, Fig. 5A). The high median *pN/pS* of populations from the deeper layers suggested relaxed selection pressure relative to populations from shallower layers, which was in agreement with previous results (12). The general explanation for such a trend is a slower replication rate, which leads to smaller effective population sizes and a reduced selection effect in the deep ocean (11, 12). However, further work is needed to substantiate this hypothesis.

Since transposases and deep-sea ORFs were both under relaxed selection, we tested whether high transposase abundance correlated with relaxed selection in a population. However, we did not observe a correlation between the median *pN/pS* of MAGs and their

**TABLE 1** Multiple stepwise regression for log-transformed transposase abundance (%) of 1,402 MAGs[a]

| Explanatory variable(s) | P for F-test | Cumulative $R^2$ |
|---|---|---|
| Depth | $<10^{-10}$ | 0.34 |
| Depth + secretory CAZyme (%) | $<10^{-10}$ | 0.37 |
| Depth + secretory peptidase (%) | $<10^{-10}$ | 0.37 |
| Depth + genome size (Mbp) | $<10^{-10}$ | 0.52 |
| Depth + taxon (class) | $<10^{-10}$ | 0.53 |
| Depth + taxon + genome size + secretory CAZyme + secretory peptidase | NA[b] | 0.62 |

[a]An ANOVA test was performed on each newly added covariate.
[b]NA, not applicable.

transposase abundance (Spearman ρ = −0.075, P = 0.08). Furthermore, the relationship between the selective pressure experienced by a population and its transposase abundance was inconsistent across different depths (Fig. 5B). It has been previously suggested that small populations would experience relaxed selection against mobile genetic elements (38, 39). However, our analyses show that although relaxed selection and transposase enrichment co-occur in deep oceans, we did not observe this trend on a genome-by-genome basis, suggesting that there is not a direct connection between transposase abundance and relaxed selection at a genome-resolved scale, and thus within individual microbial populations.

## Transposons/integrons carry a high proportion of ORFs related to defense mechanisms

One important question regarding high transposase abundance in the deep sea is whether these transposases are neutral or deleterious, or whether they perform important functions in the genomes encoding them. In some cases, transposases are deleterious and proliferate as a result of genetic drift (40), but transposases can also benefit the host by introducing functional and regulatory genes (6, 25, 41, 42).

Transposases mediate the migration and integration of cassette sequences into host genomes; a transposon or integron is the combination of a transposase and its cassettes. Thus, cassette sequences are functionally similar to auxiliary metabolic genes (AMGs) (43) in that they are carried by a selfish MGE (or viruses in the case of AMGs) to increase the fitness of the host and therefore increase the fitness of the MGE.

To determine whether transposons in marine habitats carry advantageous genes, we began by querying the functional categories of cassette sequences. The software package Integron Finder (10) was used to locate cassettes on contigs; the program searches for the two palindromic flanking sites necessary for transposase-mediated recombination as well as a nearby transposase/integrase. We treated transposons and integrons as equivalent, due to the sequence and function similarity between integrases and transposases (10). We found 8,519 cassette ORFs from the co-assembled Malaspina (17) and *Tara* Oceans (16, 18) metagenomes (see Materials and Methods). Only 27% of the cassette sequences were assigned with COG annotations, which was a lower annotation rate compared to other ORFs in the *Tara* Oceans (55%–60%) and Malaspina (68%) metagenomes.

Compared to the rest of the metagenome, cassettes were enriched in ORFs of the COG categories "replication, recombination, and repair," "defense mechanisms," and "mobilome: prophages, transposons" (Fig. 6A). Transposons/integrons are expected to be enriched in ORFs related to the mobilome and replication/recombination. The prevalence of defense mechanism genes in cassettes was noteworthy: defense mechanisms accounted for 14.3% of known function calls in cassettes, but only accounted for 2.47% of known function calls in non-cassette ORFs ("known" was defined as all functional groups except "function unknown" and "general function prediction"). Out of the 276 defense mechanism cassettes, 123 of them encoded toxin/antitoxin genes. Since transposases are more pervasive on particles, toxin and defense genes they carry would

be useful in competition and protection on crowded particles (44–47). Moreover, the percentage of secretory CAZymes and peptidases in metagenomes both correlated with the abundance of defense mechanism ORFs (Fig. 6B). Similarly, we also found a strong correlation between the gene abundance of defense mechanism ORFs and that of biofilm-associated ORFs (Fig. S6). Thus, if defense mechanism genes in the deep ocean were more highly expressed, this would substantiate the hypothesis that deep ocean microorganisms benefit from novel genes introduced by integrons/transposons.

## ORFs related to defense mechanisms, secretory CAZymes, and biofilm-associated genes have higher expression in the deep sea

To determine whether defense mechanism genes are more highly expressed and how this correlates to expression of particle-associated genes, we calculated their RNA/DNA ratios (defined as the transcript abundance of a target gene divided by its gene abundance) of secretory CAZyme, secretory peptidase, defense mechanism, and transposase ORFs in each *Tara* Oceans sample with a paired metagenome and metatranscriptome ($n = 94$). Malaspina samples were excluded from this analysis because no metatranscriptomes were sequenced.

The ORFs related to secretory CAZymes and defense mechanisms had greater RNA/DNA ratios in the mesopelagic zone than in the surface and DCM (Fig. 7), demonstrating a greater need for these genes as deep ocean microbial communities switch toward a particle-associated lifestyle. The RNA/DNA ratio of biofilm-associated ORFs also supported this switch to particle-associated lifestyle (Fig. S7). In contrast, although transposases were more abundant in the mesopelagic zone (Fig. S8), their RNA/DNA ratios were similar across all depths (Fig. 7). A possible explanation is that although transposases are not upregulated, they sometimes carry beneficial cassette genes, which are increasingly expressed in deep-sea microbial genomes.

## Conclusion

Our analysis provides insights into the factors driving the increasing abundance of transposases with ocean depth, highlighting the interplay between the particle-associated lifestyle, expanded genome size, and selection for defense mechanism genes (Fig. 8). We hypothesize that as microbial communities shift from a predominantly planktonic to particle-associated lifestyle from the surface to the bathypelagic zone, microbial communities become more densely packed on particles, leading to rampant transposase spread and more intense resource competition. Additionally, particle-associated microorganisms tend to have larger genomes, which are associated with high transposase abundance. These high abundances of mobile genetic elements actively shape the ecology of particle-associated microbial communities by introducing novel genes, with cassette sequences being particularly enriched in defense mechanism genes. These genes are highly expressed in the deep ocean, offering competitive advantages in the competitive particle-associated environment (44, 46).

When interpreting these results, it is important to consider certain caveats. Our analysis focused on pelagic ocean water samples to minimize potential confounding variables, and thus, these observed trends may not apply to regions with unique characteristics, such as Arctic sea ice (48) and deep-sea hydrothermal vents (14). Moreover, as our results are based on correlative analyses, further experimental evidence is needed to establish causation and to identify mechanistic relationships between these variables.

Understanding these trends in mobile genetic element abundance is important for understanding gene flow, competition, and other ecological characteristics of the dark ocean, one of the largest habitats on Earth. Our results show a strong association between transposase abundance and a particle-associated lifestyle, suggesting that transposons may enable microbial lineages to compete in crowded biofilm-associated habitats. We did not find any correlation between the strength of selection and transposase abundance at the genome level, indicating that relaxed selection pressure is

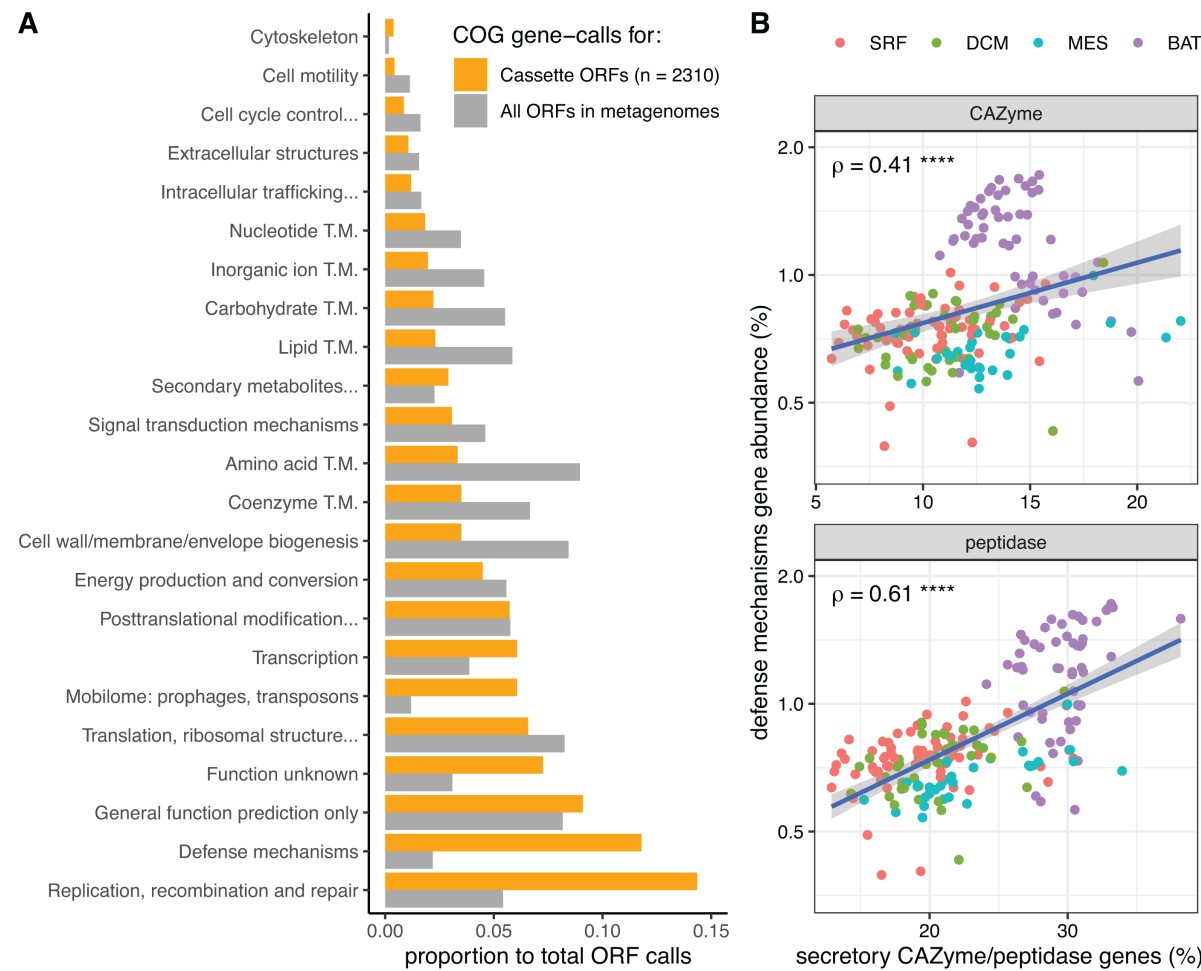

**FIG 6** Cassette sequences include high proportions of defense mechanism genes, which are more abundant in microbial communities that rely heavily on a particle-associated lifestyle. (A) The distribution of COG functional categories in cassette and non-cassette metagenome ORFs. T.M., transport and metabolism. One thousand nine hundred fifty-one cassette ORFs with COG function calls were identified from the 10 co-assembled *Tara* metagenomes, and 359 from the co-assembled Malaspina metagenome. Five million ORFs with COG function calls were then sampled from the *Tara* Oceans and Malaspina metagenomes according to the ratio of identified cassettes described above. (B) The correlation between the abundance of secretory CAZyme/peptidase ORFs and defense mechanism ORFs in metagenomes. Samples from different depths are distinguished by different colors. The Spearman's correlation coefficients, ρ, are shown on top left. **** indicates that $P < 0.0001$.

not the primary driver for the high transposase abundance in the deep ocean. Overall, our results suggest an emerging understanding of the ocean as a stratified system in which the deep ocean acts as a gene-sharing highway, fostering networks of gene exchange, particularly on particles. This results in large genomes with many defense-oriented genes in the deep-sea microbial communities, contrasting with the streamlined and specialized genomes that dominate the surface oceans. Future experimental studies should establish mechanistic connections between transposase gene cassette contents and microbial activity, particularly in planktonic and particle-associated communities.

## MATERIALS AND METHODS

### Analysis of the ocean microbial reference catalog v2 (OM-RGC.v2) from the *Tara* Oceans project

OM-RGC.v2 contained 47 million non-redundant ORFs (15, 16), and the coverage of each ORF in every metagenomic ($n = 138$) and metatranscriptomic ($n = 152$) sample ([https://](https://)

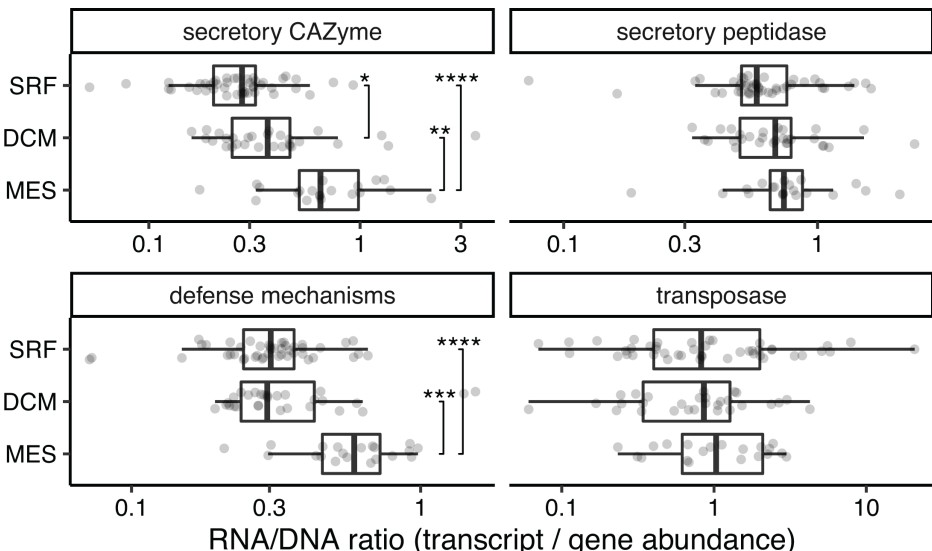

**FIG 7** Particle-associated and defense mechanism ORFs are more highly expressed in deeper waters, but transposases are not. Log-transformed RNA/DNA ratios of target genes in each sample, separated by depth. *$P < 0.05$, **$P < 0.01$, ***$P < 0.001$.

www.ocean-microbiome.org/) (15). Samples from the Arctic Ocean were excluded. We identified transposases and biofilm-associated ORFs in OM-RGC.v2 through TBLASTN (e-value $<10^{-5}$). The transposase database contained "transposase" and "integrase" genes from the Pfam database (49) (Table S1). We identified ORFs of the "defense mechanisms" category through COG annotations in OM-RGC.v2. Finally, the coverage data in OM-RGC.v2 was used to determine the relative abundance of a target gene in a sample.

## Acquisition of global ocean metagenomes

Metagenomic reads of the *Tara* Oceans Project and Malaspina Expedition were downloaded from the European Nucleotide Archive under study accessions PRJEB402 and PRJEB44456, respectively. Methods for sample collection and Illumina sequencing were described by Sunagawa et al. (50) for the *Tara* Oceans Project and Acinas et al. (17) for the Malaspina Expedition. For the assembled contigs used as references for mapping of the *Tara* Oceans metagenomes, we used 10 existing co-assembled metagenomes from the 10 oceanic provinces (18). For the reference contigs of the Malaspina metagenomes, we downloaded the co-assembled metagenome of all 58 samples (17). Samples from the *Tara* Oceans were filtered across a variety of size fractions. For full metagenomic analysis, we examined only samples that were filtered through 1.6 µm or 3 µm filters and collected on 0.22 µm filters (50). Samples from the Malaspina Expedition were collected from the 0.2 µm–0.8 µm and 0.8 µm–5 µm size fractions (17). We used EukDetect (51) to confirm that samples collected with a larger size fraction did not have a higher contribution from eukaryotic reads (see Supplementary Text).

## Prediction of secretory CAZyme and peptidase ORFs

We adapted the methods employed in Zhao et al. (21) to assess whether microbial communities were largely planktonic or particle-associated by quantifying the relative abundance of secretory CAZymes and peptidases. Predicted CAZymes and peptidases were annotated using DIAMOND (2.0.15) (52) BLASTP (e-value $<10^{-10}$) to search against the dbCAN (53) and MEROPS (54) databases, respectively. SignalP (5.0) (55) was used to identify signal peptides. We used the Gram-positive mode for ORFs affiliated with *Actinobacteria* and *Firmicutes*, Gram-negative mode for other bacterial phylum, and Archaea mode for ORFs affiliated with the *Archaea* domain. Only ORFs in the *Bacteria* or *Archaea* domains were included in the analysis. For MAGs, counts of secretory CAZymes

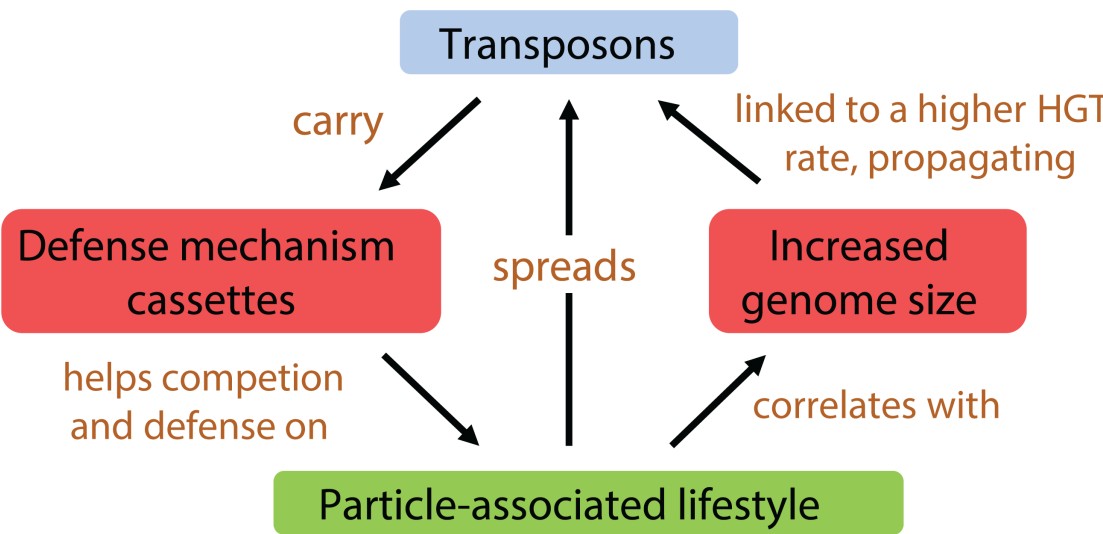

**FIG 8** Potential mechanisms linking a particle-associated lifestyle with the high abundance of transposons in the deep ocean.

and peptidases were used instead of abundance; validation based on predicted lifestyles of known taxa confirmed that this gave an accurate prediction of lifestyle (see Supplementary Text).

## MAG selection for size fraction and lifestyle analysis

To analyze transposase abundance by size fraction (and thus lifestyle), we used a collection of 1,888 bacterial and archaeal MAGs derived from *Tara* Oceans co-assemblies recovered by Delmont et al. (20). All of these MAGs exhibited >70% completion, with an average completion of 87.1% and an average redundancy of 2.5%. Previous analysis of these MAGs used mapping to determine the relative abundances of each MAG across different size fractions. Bins were only considered to be present if ≥25% of the length of the MAG was mapped by reads in a given sample (20). We took the size fraction with the highest proportion of mapped reads to be the designated size fraction of that MAG. Four MAGs could not be assigned a size fraction and were thus left out, leaving a final sample size of 1,884. A large proportion of our MAGs were assigned the smallest size fraction ($n$ = 1,586), with fewer in the larger size fractions ($n$ = 175 for 5 µm–20 µm, $n$ = 64 for 20 µm–180 µm, $n$ = 59 for 180 µm–2,000 µm).

To quantify transposase abundance, we used BLASTP (56) to search each MAG against the transposase database previously discussed. The percent transposase abundance was calculated as the number of unique BLASTP hits (e < $10^{-5}$) divided by the total number of ORFs in the MAG, as identified by Prokka (v. 1.14.6) (57) using default settings.

All data processing and graphics for this analysis was done in R (v.4.3.2) (58) using the tidyverse ecosystem (v.2.0.0) (59).

## MAG selection for depth, genome size, and taxonomy analysis

To conduct genome-resolved analyses focused on depth, genome size, taxonomy and signatures of selection, we selected 1,147 out of 2,631 MAGs that had previously been recovered from the *Tara* Oceans metagenomes (18) and 255 out of 317 MAGs recovered from the Malaspina metagenomes (17). All selected MAGs had <5% of eukaryotic sequences(in base pairs, Tiara [60] was used to identify putative eukaryotic contigs), completeness >70%, and redundancy <10% (both criteria were calculated with CheckM [61]).

## Determination of depth and lifestyles for each MAG

MAGs that originated from co-assemblies in a specific province in the *Tara* Oceans data set (e.g., Red Sea), as in Tully et al. (18), were used to recruit reads from all *Tara* Oceans samples ($n = 180$) using Bowtie2 (default parameters). Each province was performed separately. SAM files were converted to BAM files using samtools (62) and used to determine RPKM (reads per kilobase pair MAG per million pair metagenome) as in Graham et al. (63).

To assign the depth origin of a MAG from the *Tara* Oceans metagenomes, we summed up its RPKM in surface, DCM, and mesopelagic samples (three groups, separately). The layer with the highest RPKM sum was assigned as the depth of that MAG. All Malaspina MAGs belonged to the bathypelagic layer.

## Mapping and calculating *pN/pS* ratios of MAGs

To assess the strength of selection of target ORFs within specific MAGs, we calculated the *pN/pS* ratio (the proportion of nonsynonymous mutations over the proportion of synonymous mutations) for all ORFs within each target MAG. To do this, we mapped the metagenomic reads against the assembled contigs for each metagenome. We mapped the raw reads of each of the *Tara* Oceans metagenomes to the co-assembly from their corresponding provinces, and all raw reads of each of the Malaspina metagenomes to one co-assembled metagenome using Bowtie2 (v2.2.9; paired-end alignment with default parameters) (64). We calculated *pN/pS* for each ORF within each individual MAG using anvi'o (65) with the script "anvi-script-calculate-pn-ps-ratio" with the "--min-coverage" flag set to 20. The *pN/pS* ratios were calculated on a sample-per-sample basis, so an ORF from a co-assembled metagenome might have multiple *pN/pS* ratios from different samples.

## Integron and cassette sequences detection

We used Integron Finder (v2) (10) to identify cassette sequences from the *Tara* Oceans and Malaspina metagenomes. Integron Finder uses HMMER to locate the integron-integrase *intI*, which is conserved for most integrons. Then, cassette sequences are identified with the near-palindromic flaking regions. anvi'o (65) was used to identify and annotate ORFs on metagenomic contigs using the script "anvi-run-ncbi-cogs." If the anvi'o ORF calls from Prodigal (66) were within 100 bps (start and stop) of the Integron Finder's cassette calls, the anvi'o COG annotations were used for those cassette ORFs.

## Statistical analysis

We used the Wilcoxon signed-rank test to compare the differences between means of two numeric variables, and the Spearman's rank correlation was used to determine the association between two numeric variables. Multiple hypothesis *P*-values were adjusted using the Benjamini-Hochberg procedure. All linear regressions had a normal distribution of residuals (Shapiro-Wilk test). An ANOVA *F*-test was used to perform model comparisons (whether adding additional variables makes the prediction significantly better). Statistical significance was assumed if $P < 0.05$. All statistical analyses were performed using base R packages (v.4.1.3 and v.4.3.2).

### ACKNOWLEDGMENTS

We would like to thank Dr. Shinichi Sunagawa for kindly providing information about the *Tara* Oceans data sets and the Ocean Microbial Reference Catalog, and Dr. Silvia G. Acinas for providing information about the Malaspina Deep Ocean data set. Dr. Murat Eren provided help with anvi'o, Mike Tie provided assistance with server administration, and Mark McKone provided statistical advice.

Funding for J.Z., O.K., and T.O. was provided by grants from the Towsley Endowment at Carleton College. Funding for T.O. was also provided by the Rosenow fund at Carleton

College, and funding for T.D.R.H. was provided by the Summer Science Fellows program at Carleton College.

## AUTHOR AFFILIATIONS

[1]Carleton College, Northfield, Minnesota, USA
[2]Department of Medicine, Washington University in St. Louis, St. Louis, Missouri, USA
[3]Department of Molecular Biology, Cell Biology and Biochemistry, Brown University, Providence, Rhode Island, USA
[4]Yale School of Medicine, Yale University, New Haven, Connecticut, USA
[5]Marine & Environmental Biology, Department of Biological Sciences, University of Southern California, Los Angeles, California, USA

## AUTHOR ORCIDs

Benjamin J. Tully  http://orcid.org/0000-0002-9384-7635
Rika E. Anderson  http://orcid.org/0000-0001-5946-7922

## FUNDING

| Funder | Grant(s) | Author(s) |
|---|---|---|
| Towsley Endowment, Carleton College | | Juntao Zhong |
| | | Troy Osborn |
| | | Oleksandr Kyrysyuk |
| Rosenow Fund, Carleton College | | Troy Osborn |
| Summer Science Fellows Program, Carleton College | | Thais Del Rosario Hernández |

## AUTHOR CONTRIBUTIONS

Juntao Zhong, Conceptualization, Formal analysis, Methodology, Visualization, Writing – original draft, Writing – review and editing | Troy Osborn, Formal analysis, Visualization, Writing – original draft, Writing – review and editing | Thais Del Rosario Hernández, Formal analysis, Investigation, Writing – review and editing | Oleksandr Kyrysyuk, Formal analysis, Investigation, Writing – review and editing | Benjamin J. Tully, Data curation, Resources, Writing – review and editing | Rika E. Anderson, Conceptualization, Formal analysis, Funding acquisition, Methodology, Project administration, Supervision, Writing – original draft, Writing – review and editing

## CODE AND DATA AVAILABILITY.

All Python, R scripts, code explanations, and raw data for analysis are publicly accessible on GitHub at https://github.com/carleton-spacehogs/transposase-deep-ocean.

## ADDITIONAL FILES

The following material is available online.

### Supplemental Material

**Supplemental Information (mSystems00067-24-s0001.docx).** Supplemental figures and tables.
**Table S1 (mSystems00067-24-s0002.xlsx).** Accession numbers of 2,307 seed sequences of "transposase" and "integrase" genes from the Pfam database.

## Open Peer Review

**PEER REVIEW HISTORY (review-history.pdf).** An accounting of the reviewer comments and feedback.

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
