## [Reviewer comments · mSystems]

Increasing transposase abundance with ocean depth correlates with a particle-associated lifestyle

Juntao Zhong, Troy Osborn, Thais Del Rosario Hernandez, Oleksandr Kyrtsyuk, Benjamin Tully, and Rika Anderson

Corresponding Author(s): Rika Anderson, Carleton College

Review Timeline:

Submission Date:

January 15, 2024

Accepted:

January 25, 2024

Editor: Samuel Chaffron

Reviewer(s): Disclosure of reviewer identity is with reference to reviewer comments included in decision letter(s). The following individuals involved in review of your submission have agreed to reveal their identity: Tom Delmont (Reviewer #2)

Transaction Report:

DOI: <https://doi.org/10.1128/msystems.00067-24>

Re: mSystems00067-24 (Increasing transposase abundance with ocean depth correlates with a particle-associated lifestyle)

Dear Dr. Rika Anderson:

Congratulations, your manuscript has been accepted, and it will be forwarded to the ASM production staff for publication. Nevertheless, as suggested by Reviewer #2, you may follow its recommendations for improving Figure 3.

Your paper will first be checked to make sure all elements meet the technical requirements. ASM staff will contact you if anything needs to be revised before copyediting and production can begin. Otherwise, you will be notified when your proofs are ready to be viewed.

Please provide in the "Data Availability" paragraph additional information regarding the data description. Accession numbers and/or URLs for accessing the sequence data need to be specified in this section.

Publication Fees: For information on publication fees and which article types have charges, please visit our website. We have partnered with Copyright Clearance Center (CCC) to collect author charges. If fees apply to your paper, you will receive a message from no-reply@copyright.com with further instructions. For questions related to paying charges through RightsLink, please contact CCC at ASM_Support@copyright.com or toll-free at +1-877-622-5543. CCC makes every attempt to respond to all emails within 24 hours.

Featured Image Submissions: If you would like to submit a potential Featured Image, please email a file and a short legend to msystems@asmusa.org. Please note that we can only consider images that (i) the authors created or own and (ii) have not been previously published. By submitting, you agree that the image can be used under the same terms as the published article. Image File requirements: TIF/EPS, 7.5 inches wide by 8.25 inches tall (at least 2,250 pixels wide by 2,475 pixels tall), minimum 300 dpi resolution (600 dpi preferred), RGB, and no figure elements, e.g., arrows or panel labels. The legend should be a short description of the image, 1-2 sentences recommended.

Sincerely,
Samuel Chaffron
Editor
mSystems

Reviewer #2 (Comments for the Author):

Authors have adequately addressed my main concern regarding the Tara Oceans size fractions. I am very pleased to learn from the study that "transposases are indeed enriched in genomes that are enriched in larger size fractions and can thus be considered particle associated, and the difference is statistically significant". Results are very clear and add a critical layer of information supporting the main conclusions of the study.

Maybe, for the Figure 3, authors could consider adding boxplots for the length of the MAGs, as it might show higher density of transposases over A LONGUER genomic length. This is one advantage of working with genomics compared to metagenomics. However, if the results are not clear authors do not have to follow this suggestion. I just feel this figure could be slightly improved. Another suggestion would be to add bar plots showing the relative proportion of major phyla for each box plot, so that the reader can link increased density of transposases to the shift in taxonomy between size fractions. Again, merely a suggestion here.

Best regards

Tom Delmont